

# Relationships between crayfish population genetic diversity, species richness, and abundance within impounded and unimpounded streams in Alabama, USA

Zanethia C. Barnett[1] and Ryan C. Garrick[2]

[1] Southern Research Station, USDA Forest Service, Oxford, Mississippi, United States
[2] Department of Biology, University of Mississippi, University, Mississippi, United States

Corresponding author
Zanethia C. Barnett,
zanethia.c.barnett@usda.gov

## ABSTRACT

Understanding the relationship between multi-scale processes driving community- and population-level diversity can guide conservation efforts. While the importance of population-level genetic diversity is widely recognized, it is not always assessed for conservation planning, and positive correlations with community-level diversity are sometimes assumed, such that only the latter is measured. We surveyed species richness and cumulative multispecies abundance of crayfishes in impounded and unimpounded streams in the southern Appalachian Mountains (Alabama, USA). We simultaneously assessed levels of population genetic diversity within two focal crayfishes (*Faxonius validus* and *F. erichsonianus*) using nuclear (nDNA; inter-simple sequence repeat (ISSR)) and mitochondrial DNA (mtDNA; mitochondrial DNA cytochrome oxidase subunit I (mtCOI)) markers. We then tested for species-genetic diversity correlations (SGDCs), species diversity-abundance correlations (*i.e.*, more individuals hypothesis, MIH), and abundance-genetic diversity correlations (AGDCs) across sites. We also examined the relationship between each of the three different types of correlation (*i.e.*, species richness, cumulative multispecies abundance, and population genetic diversity) and stream habitat characteristics and fragmentation. Surprisingly, based on *F. validus* mtDNA data, sites with the greatest multispecies abundance had the lowest genetic diversity, indicating a negative AGDC. However, no AGDC was evident from nDNA. There was no evidence of SGDCs for *F. validus* based on either of the two genetic data types. For *F. erichsonianus*, there was no evidence for SGDC or AGDC. When considering the community-level data only, there was no support for the MIH. Stream width was positively correlated with *F. validus* genetic diversity, but negatively correlated with multispecies abundance. Similarly, species richness was positively correlated with stream width in unimpounded streams but negatively correlated with width in impounded streams. These findings indicate that community-level diversity cannot be indiscriminately used as a proxy for population-level diversity without empirically testing this correlation on the focal group. As such, community- and population-level assessments for multiple crayfish species are needed to better understand drivers of diversity and eco-evolutionary processes which will aid in the conservation of this vulnerable taxonomic group.

# INTRODUCTION

## Parallel processes may structure biodiversity

Community-level taxonomic diversity (*e.g.*, species richness) and population-level genetic diversity are both key components of biological diversity (*Allendorf & Luikart, 2007*). According to the theory of island biogeography, the balance between colonization and local extinction determines species richness at a given site (*MacArthur & Wilson, 1967*), whereas the counteracting forces of gene flow and drift determine the standing levels of population genetic variation (*Wright, 1940*; *Kimura, 1983*; *Nei, 1987*). *Velend (2005)* proposed that if parallel processes operate at both the community- and population-level, this should result in a positive species-genetic diversity correlation (SGDC). In addition to SGDCs, it has also been proposed that species richness and/or population-level genetic diversity may be positively correlated with the cumulative multispecies abundance of individuals within a community (*Lamy et al., 2017*). There are several reasons why this can occur. First, under the more individuals hypothesis (MIH; *Storch, Bohdalková & Okie, 2018*), community-level species richness at a local site may be high if populations of these species are large and stable in size, such that local extinction driven by negative feedback between intrinsic stochastic ecological and genetic processes (*e.g.*, Allee effects, genetic drift, and inbreeding) is negligible. Likewise, direct and indirect species interactions can be beneficial (*e.g.*, mutualism or facilitation), and species-rich communities may also be characterized by greater redundancy among key role players that are critical to ecosystem functioning, making these communities more resilient to environmental perturbations (*Waide et al., 1999*; *Finke & Snyder, 2008*; *Lamy et al., 2017*). Second, according to the abundance-genetic diversity correlation (AGDC) hypothesis (*Johansson et al., 2005*; *Overcast, Emerson & Hickerson, 2019*), if effective population size ($N_e$) and census population size ($N_c$) scale with one another (which is supported by a meta-analyses (*Frankham, 1996*; *McCusker & Bentzen, 2010*)), then environmental conditions that promote high cumulative multispecies abundance of individuals should also translate into relatively weak effects of drift, preventing the loss of allelic diversity (*Storch, Bohdalková & Okie, 2018*). Large $N_e$ also provides the basis for more efficient natural selection, and balancing selection may play an important role in retaining genetic variation within populations (*Chesson, 2000*).

Numerous studies have assessed evidence for positive SGDCs, MIH, and AGDCs in diverse groups of organisms (*Hurlbert, 2004*; *Dudgeon & Ovenden, 2015*; *Xie et al., 2021*; *Bucholz et al., 2023*), but negative and no correlation have also been detected when factors such as habitat fragmentation and disturbance affect species richness, population genetic diversity, and multispecies abundance differently (*Scribner et al., 2001*; *Johansson et al., 2005*; *Wei & Jiang, 2012*; *Šímová, Li & Storch, 2013*; *Seymour et al., 2016*; *Watanabe &*

*Monaghan, 2017*; *Reisch & Schmid, 2019*). Recent meta-analyses (*Xie et al., 2021*) and reviews of published studies that investigated evidence for SGDCs (*Lamy et al., 2017*) and MIH (*Storch, Bohdalková & Okie, 2018*) found that most studies (80%-SGDC and 72%-MIH) reported positive correlations. Nonetheless, no relationship between species richness, genetic diversity, and abundance was detected in studies conducted in highly disturbed areas or along environmental gradients (*Carnicer & Díaz-Delgado, 2008*; *Wei & Jiang, 2012*; *Šímová, Li & Storch, 2013*; *Fan et al., 2021*; *Petersen et al., 2022*). Additionally, negative SGDCs have also been documented when there is an opposite influence of environmental drivers (*e.g.*, altitude, temperature) and/or competition on species diversity, genetic diversity, and abundance (*Scribner et al., 2001*; *Currie et al., 2004*; *Seymour et al., 2016*; *Lamy et al., 2017*; *Marchesini et al., 2018*; *Ishii et al., 2022*). Taken together, these inconsistent results indicate that community- and population-level processes may not operate in parallel ways in some ecological contexts and groups of organisms (*Lamy et al., 2013*; *Storch, Bohdalková & Okie, 2018*).

SGDC, MIH, and AGDC are not mutually exclusive, but there are several reasons why only a subset (or perhaps just one) of these hypotheses may receive support in a given study. First, extrinsic factors may impact species differently (*Kahilainen, Puurtinen & Kotiaho, 2014*; *Bucholz et al., 2023*). This may be due to divergent ecological functions, life histories, ecological optima, phenotypic plasticity and/or capacity to respond to dynamic environmental conditions. Second, contrasting carrying capacities of habitats that support local communities may impose constraints on diversity and abundance of some species but not others (*Loreau, 2000*). Additionally, rare and specialized species' genetic diversity often have little correlation to carrying capacities of habitats, while in most communities species diversity and abundance are strongly correlated with carrying capacities (*Velland, 2005*).

While the assessment of SGDC, MIH, and AGDC requires a field- and labor-intensive multi-level approach aimed at investigating different levels of biodiversity, as well as their evolutionary ecological drivers within a community, such studies can provide important insights into the "scaling" of processes shaping biodiversity, which in turn have practical implications for conservation (*Kahilainen, Puurtinen & Kotiaho, 2014*). For example, if all three hypotheses are supported in a given study, then natural resource managers could collect any one of the three types of diversity data (*i.e.*, species richness, genetic diversity, or cumulative multispecies abundance) and make reasonable predictions about the other two (*Kahilainen, Puurtinen & Kotiaho, 2014*; *Overcast, Emerson & Hickerson, 2019*). Likewise, if two of the three hypotheses were supported, this information could be used to identify which one of the three data types should be prioritized in biodiversity assessments (*e.g.*, if both SGDC and MIH are true, then species richness data alone could be used to predict genetic diversity and abundance; *Bucholz et al., 2023*). Although the robustness of such extrapolations should be empirically verified at several sampling sites, in such situations there is potential for this predictive framework to improve the cost-effectiveness of biodiversity monitoring and conservation strategies. Furthermore, making assumptions without assessing biodiversity correlations can lead to sub-optimal or detrimental conservation strategies.

## Streams as study systems for assessing parallel processes

When assessing the correlations between biodiversity metrics, having a system that is explicitly linked through dispersal takes into account the spatial effects of migration and dispersal which are key processes involved in diversity dynamics (*Vellend & Geber, 2005*; *Altermatt, 2013*; *Seymour et al., 2016*). As such, a system, such as streams, with a dendritic network, provide a suitable and tractable study system for empirically testing whether parallel processes, operating at hierarchically nested levels, structure biodiversity in similar ways. This is because stream boundaries are clearly demarcated and environmental characteristics (*e.g.*, habitat area and substrate size), and distributions of stream-dependent biota, are usually predictably structured along an upstream-downstream gradient (*Vannote et al., 1980*; *Rimalova, Douda & Stambergova, 2014*; *Barnett et al., 2022*). Furthermore, because most stream-dependent biota are regionally constrained by the network spatial arrangement (*Grant, Lowe & Fagan, 2007*) and locally restricted to the stream channel, it is possible to temporarily isolate sections so they can be exhaustively sampled (*e.g.*, *via* block net multi-pass electrofishing). This provides an otherwise rare opportunity for quantitative assessments of stream community species richness and abundance (*Hornbach & Deneka, 1996*; *Ode, Rehn & May, 2005*; *Hanks, Kanno & Rash, 2018*; *Barnett et al., 2020*).

## Potential impacts of stream fragmentation

Notwithstanding the aforementioned advantages of streams as study systems, loss of connectivity due to human-mediated habitat fragmentation (*e.g.*, dams and impoundments) is common, and this can impact biodiversity of resident biota by altering stream habitat, community composition, and population genetic structure (*Barnett et al., 2020*, *2022*, *2023*). Dams and impoundments are among the most prevalent and extreme alteration on fluvial systems (*The Heinz Center, 2002*; *Liermann et al., 2012*; *Grill et al., 2015*), with river fragmentation and flow regulation being one of the largest biological effects of dams (*Stanford & Ward, 2001*; *Grill et al., 2015*; *Barnett et al., 2021*). Unlike unimpounded streams where organisms experience the natural flow variability and can freely move throughout the stream system, organisms in impounded streams are often isolated to one stream section and natural flow variability is greatly reduced causing changes to habitat composition and accessibility. These changes can impact species richness, abundance, and dispersal throughout the stream system. Furthermore, habitat fragmentation is expected to reduce within-patch species richness and abundance due to decreases in habitat complexity which can reduce suitable habitat for habitat-specialists (*Barnett et al., 2022*), as well as reduce intraspecific genetic diversity due to restricted dispersal and gene flow leading to isolation and drift (*Vellend & Geber, 2005*; *Hartfield, 2010*; *Barnett et al., 2020*). However, a decoupling of the effects on these two diversity metrics may occur when habitat fragmentation impacts the dispersal ability of some species differently than others (*Lamy et al., 2017*). For example, crayfish that prefer smaller streams and naturally disperse upstream, up steep slopes and against fast water velocities, may be capable of bidirectional gene flow within fragmented streams, while those preferring larger sized streams may have unidirectional downstream or no gene flow

between fragmented sections (*Hartfield, 2010*; *Barnett et al., 2020*). Thus, whether changes to species richness, abundance, and population genetic diversity in fragmented systems mimic those in connected systems is an important question for conservation biologists.

## Crayfish as focal group of organisms

Nearly 70% of the world's freshwater crayfish species are found in the United States (US) (*Crandall & Buhay, 2008*; *Richman et al., 2015*), with the southeastern US being the major center of diversity (*Hobbs, 1989*; *Richman et al., 2015*). These organisms play fundamental roles in stream ecosystem trophic processes (*e.g.*, processing detritus, altering the composition of macrophyte and substrate, transferring energy to higher level organisms), and they are often considered ecosystem engineers due to their modification of the physical habitat (*e.g.*, creating habitat for other organisms through burrow creation, bedform alterations in streams, *etc.*) (*Momot, 1995*; *Usio, 2000*; *Statzner, Peltret & Tomanova, 2003*; *Usio & Townsend, 2004*; *Reynolds, Souty-Grosset & Richardson, 2013*; *Krupa, Hopper & Nguyen, 2021*). Alarmingly, 48% of North American crayfish species are threatened (*Taylor et al., 2007*), with extinction rates likely to increase by more than an order of magnitude over the next several decades (*Ricciardi & Rasmussen, 1999*; *Cowie, Bouchet & Fontaine, 2022*; *Finn, Grattarola & Pincheira-Donoso, 2023*). Hence, there is an immediate need for effective crayfish conservation strategies (*e.g.*, *Taylor et al., 2019*). However, conservation planning at the community-level has been emphasized much more strongly than population-level genetic diversity, as has preservation of specific "units" or phenotypes over the evolutionary processes that generate this diversity (*e.g.*, large self-sustaining populations living in heterogeneous landscapes; *Moritz, 2002*). Thus, to effectively conserve crayfish diversity, understanding the extent to which similar processes structure biodiversity across different levels of biological organization is key.

In the present article, we assessed evidence for parallel processes, operating at hierarchically nested levels, within connected and fragmented (*i.e.*, unimpounded *versus* impounded) streams in the southern Appalachian Mountains, Alabama. SGDC and AGDC were each tested using *Faxonius validus* and *F. erichsonianus*. These were chosen as our focal species because they share many ecological traits (*e.g.*, life span, mating season, burrowing capabilities, preferred habitat) but differ in their preferred stream size and geographic range (*Bouchard, 1972*; *Williams & Bivens, 2001*; *Hopper, Huryn & Schuster, 2012*; *Barnett et al., 2020*). *Faxonius validus* occurs in small intermittent to medium-sized perennial streams in northern Alabama and southern Tennessee (*Cooper & Hobbs, 1980*; *Hobbs, 1989*), while *F. erichsonianus* occurs in medium to large streams in six southeastern states, from Mississippi to Virginia (*Hobbs, 1981*). For these species, population genetic diversity was measured using several alternative metrics based on mitochondrial DNA sequences (mitochondrial DNA cytochrome oxidase subunit I (mtCOI)), as well as complementary nuclear genetic maker (inter-simple sequence repeat (ISSR)) data. Testing of all three hypotheses (*i.e.*, including MIH) incorporated data from community-level surveys of crayfish species richness and cumulative multispecies abundance in five streams spanning two drainages. To clarify the extent to which habitat fragmentation and environmental characteristics may contribute to such correlations, we also investigated

**Table 1 Mechanisms impacting diversity at multiple scales, including environmental factors and habitat connectivity.** Our predicted outcomes of each individual factor on species richness, multispecies abundance, and population genetic diversity are denoted as being positive or negative. As such, higher species richness will be found at sites with higher wetted widths, and species richness will be lower in fragmented streams. % vegetation = percent aquatic vegetation; D50 = median substrate size; LWD = number of pieces of large woody debris.

| Mechanisms influencing diversity | Biodiversity hypotheses | | |
|---|---|---|---|
| Spatial and environmental effects | Species richness | Multispecies abundance | Genetic diversity |
| Habitat area ("wetted width") | + | + | + |
| Habitat heterogeneity ("% vegetation", "D50", "LWD") | + | + | + |
| Connectivity | | | |
| Connected ("unimpounded") | + | + | + |
| Fragmented ("impounded") | − | − | − |

whether stream habitat characteristics (size, connectivity, and habitat complexity) were correlated with species richness, abundance, and population genetic diversity (Table 1). Taken together, outcomes from these analyses should inform strategies for crayfish conservation in a geographic region of high endemism.

# MATERIALS AND METHODS

## Study area description

This study focused on lotic sections of two unimpounded and three impounded streams that are distributed across two drainages with diverse aquatic communities and numerous imperiled species (*Allen, 2001*; *McGregor & Garner, 2003*; *Phillips & Johnston, 2004*; *Barnett et al., 2022*). In the Bear Creek drainage (Tennessee River Basin), we surveyed one unimpounded (Rock Creek) and two impounded (Little Bear and Cedar creeks) streams (mean stream length: 61.5 km). In the Cahaba River drainage (Mobile River Basin), we surveyed one unimpounded (Shades Creek) and one impounded (Little Cahaba River) stream (mean stream length: 41.6 km) (Fig. 1).

Each impounded stream had one earthen storage dam (17–29 m high), creating impoundments that were 425 to 1,700 ha. The two Bear Creek drainage impoundments were installed for flood control, and water was released from outlets more than 19 m below full-pool levels from November until February and during heavy rain events. Conversely, the Cahaba River drainage impoundment stored water for municipal use. Water was released from outlets more than 10 m below full-pool levels when river water levels were insufficient to meet water municipal demands. Dams in both drainages pass normal inflows *via* spillways throughout the year.

## Site selection

In each of the three impounded streams, we selected sampling sites at set intervals (Data S1) up- and downstream of impoundments, and we mimicked the same sampling design in each of the two unimpounded streams. This approach led to selection of two to five local sites up- and downstream of impoundments, and up- and downstream of the midpoint in the unimpounded stream (hereafter, up- and downstream sections)

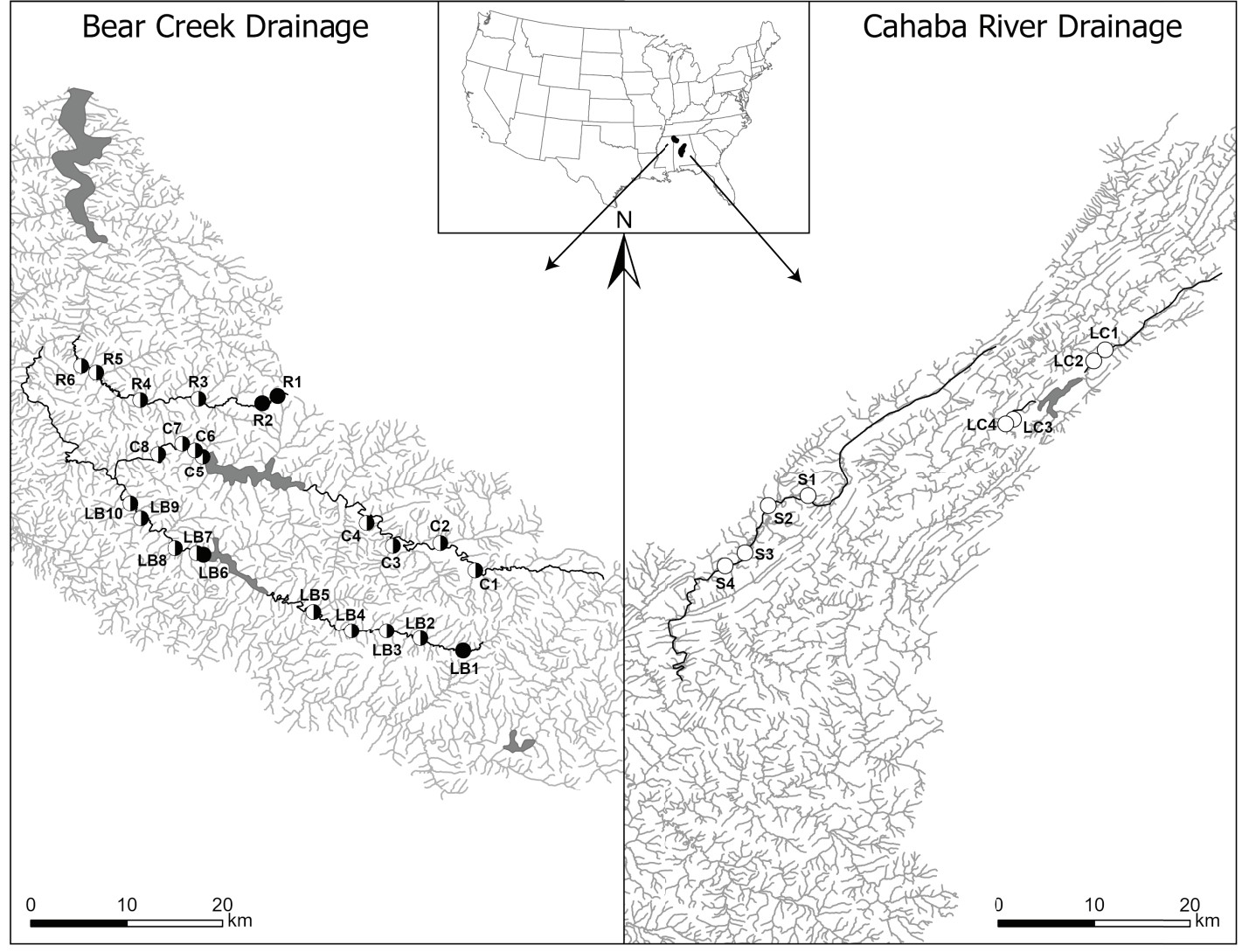

**Figure 1 Map of Bear Creek and Cahaba River drainages, Alabama, US, with shaded polygons representing impoundments and labelled circles representing collection sites.** Sites are labelled in increasing order from up- to downstream (*i.e.*, furthest upstream site = 1), with letters representing stream names (R, Rock Creek; C, Cedar Creek; LB, Little Bear Creek; S, Shades Creek; and LC, Little Cahaba River). Filled circles, sites where only *Faxonius validus* were collected; unfilled circles, sites where only *F. erichsonianus* were collected; half-filled circles, sites where both species were collected. Inset shows drainage location within the southeastern US, with the Bear Creek Drainage in the northwest corner, and the Cahaba River Drainage in the center of Alabama.

(collection data same as in *Barnett et al., 2020*). Overall, our study included 32 sites: 24 in the Bear Creek drainage (10 in Little Bear Creek, six in Rock Creek, and eight in Cedar Creek), and eight in the Cahaba River drainage (four in Shades Creek, and four in Little Cahaba River) (Fig. 1).

## Crayfish monitoring and sampling

For community-level assessment of diversity, sampling was conducted in spring (May–July) and fall (September–December) of two sampling years, during which all crayfishes were collected. During the first sampling year (spring and fall 2015), we collected

**Table 2 Mean population genetic diversity (± one standard deviation) for each of two focal crayfish species in up- and downstream sections of each stream.** $N_S$, number of sites where focal species were collected; Up, upstream; Dn, downstream; I, impounded; U, unimpounded; $h$, number of haplotypes; $h$d, haplotypic diversity; $\pi$, nucleotide diversity; PD, phylogenetic diversity; PPL, proportion of polymorphic loci.

| Focal species/Stream section and name ($N_S$) | Site codes | Stream type | No. individuals per stream section (mtCOI/ISSR) | mtCOI sequences | | | | ISSR markers PPL |
|---|---|---|---|---|---|---|---|---|
| | | | | $h$ | $h$d | $\pi$ | PD | |
| *Faxonius validus* | | | | | | | | |
| Up Little Bear (5) | LB1–5 | I | 28/24 | 5 | 0.47 (0.20) | 0.002 (0.001) | 0.004 (0.001) | 0.83 (0.16) |
| Dn Little Bear (5) | LB6–10 | I | 30/21 | 7 | 0.71 (0.06) | 0.003 (0.001) | 0.007 (0.005) | 0.86 (0.06) |
| Up Cedar (4) | C1–4 | I | 31/21 | 8 | 0.70 (0.10) | 0.004 (0.003) | 0.007 (0.004) | 0.86 (0.10) |
| Dn Cedar (4) | C5–8 | I | 21/24 | 9 | 0.76 (0.10) | 0.003 (0.001) | 0.007 (0.007) | 0.82 (0.17) |
| Up Rock (3) | R1–3 | U | 19/16 | 4 | 0.23 (0.29) | 0.001 (0.001) | 0.001 (0.002) | 0.95 (0.08) |
| Dn Rock (3) | R4–6 | U | 14/18 | 4 | 0.44 (0.50) | 0.002 (0.003) | 0.010 (0.007) | 0.94 (0.06) |
| *Faxonius erichsonianus* | | | | | | | | |
| Up Little Bear (4) | LB2–5 | I | 21/14 | 5 | 0.79 (0.20) | 0.006 (0.010) | 0.016 (0.024) | 0.82 (0.10) |
| Dn Little Bear (4) | LB7–10 | I | 23/12 | 2 | 0.23 (0.30) | <0.001 (0.001) | 0.003 (<0.001) | 0.85 (0.09) |
| Up Cedar (4) | C1–4 | I | 20/13 | 9 | 0.91 (0.06) | 0.005 (0.004) | 0.018 (0.013) | 0.72 (0.12) |
| Dn Cedar (4) | C5–8 | I | 24/16 | 7 | 0.77 (0.04) | 0.002 (0.001) | 0.019 (0.011) | 0.91 (0.11) |
| Up Rock (2) | R2–3 | U | 6/4 | 6 | 0.70 (0.10) | 0.005 (<0.001) | 0.032 (0.013) | 0.83 (<0.01) |
| Dn Rock (3) | R4–6 | U | 18/12 | 4 | 0.36 (0.40) | 0.002 (0.002) | 0.009 (0.008) | 0.95 (<0.01) |
| Up Little Cahaba (2) | LC1–2 | I | 13/8 | 6 | 0.88 (0.03) | 0.006 (0.005) | 0.028 (0.031) | 0.85 (0.03) |
| Dn Little Cahaba (2) | LC3–4 | I | 19/7 | 4 | 0.45 (0.40) | 0.001 (0.001) | <0.001 (0.001) | 0.80 (0.14) |
| Up Shades (2) | S1–2 | U | 14/8 | 5 | 0.83 (0.03) | 0.007 (0.007) | 0.027 (0.036) | 0.90 (<0.01) |
| Dn Shades (2) | S3–4 | U | 15/9 | 4 | 0.64 (0.15) | 0.001 (0.001) | 0.006 (0.003) | 0.90 (0.08) |

crayfishes from all sites in the Bear Creek drainage. During the second sampling year (fall 2016 and spring 2017), we collected crayfishes from all streams in the Cahaba River drainage. At each site, we sampled one linear reach, 30 times the wetted stream width or a minimum or maximum length of 200 to 500 m, respectively (*Barnett et al., 2022*). Stream reach lengths remained constant across seasons unless the dry season shortened a reach. Each reach was divided into two subreaches of equal length. Each subreach was simultaneously sampled by electrofishing (3–8 s/m; mean 5 s/m ± 1.2 SD) upstream subreaches and kick seining (20 plots/100 m, 2 m long × 1.5 m wide) downstream subreaches (*Barnett et al., 2021*). Because these methods are ineffective in pools and deeper waters, crayfish were collected only from riffles and runs with depths less than 1 m (≤15% of each reach). Electrofishing duration and total number of kick seines were calculated based on subreach areas. We recorded the amount of area (m$^2$) sampled by each method once the target sampling effort (electrofishing: 250–2,000 sec/subreach; kick seining: 20–50 kicks/subreach) was reached (*Barnett et al., 2021*). We used expert knowledge to identify crayfish species in the field, counting the number of individuals for each species collected at each site during each sampling round. We preserved voucher specimens (housed at Mississippi Museum of Natural Science) for each species in ≥70% ethanol and confirmed species identifications in the lab (*Hobbs, 1981*, *1989*). All collections were approved by the

State Alabama under Alabama Conservation License numbers 2016064289868680 and 2017092711268680.

For population-level assessment of genetic diversity, the two focal species, *F. erichsonianus* and *F. validus*, were both collected from the Bear Creek drainage, but only *F. erichsonianus* was collected from the Cahaba River drainage. On average, 20 individuals per species (SD = 6.6) were collected per stream section, with a total of 143 *F. validus* and 173 *F. erichsonianus* collected) (Table 2). Whole specimens were preserved in 95% ethanol.

## Characterization of population genetic diversity in two focal species

Genomic DNA was extracted from leg tissue of each *F. erichsonianus* and *F. validus* individual using a DNeasy blood and tissue kit (Qiagen, Valencia CA, USA), following manufacturer's recommendations. A 618–640 base pair (bp) region of the mitochondrial DNA cytochrome oxidase subunit I (mtCOI) gene was amplified and sequenced as described in *Barnett et al. (2020)*. Additionally, we performed genotyping using inter-simple sequence repeat (ISSR) markers (*Ziętkiewicz, Rafalski & Labuda, 1994*) for a minimum of three individuals per focal species per site (mean = 4, SD = 1.07). Whereas mtCOI is a maternally inherited haploid marker, ISSRs are generally presumed to be biparentally inherited nuclear autosomal markers, and as such, they have been used for addressing diverse questions in ecology and evolution (*e.g.*, *Wolfe, Xiang & Kephart, 1998*; *Abbot, 2001*; *Haig, Mace & Mullins, 2003*; *Dušinský et al., 2006*; *Sinn et al., 2022*). Using both nuclear and mitochondrial data could provide information on two temporal scales, with mtCOI having a smaller $N_e$ than ISSRs, potentially giving it the ability to detect more recent changes to populations (*Moore, 1995*). Together, these two types of molecular data should provide a broad overview of population genetic diversity (*Garrick, Caccone & Sunnucks, 2010*).

For initial assessment of polymorphism and scorability of ISSRs, ten primers (Data S2) were screened using a geographically representative of 10 *F. validus* and 10 *F. erichsonianus* individuals. Eight ISSR primers were designed by the author (R.C. Garrick), with the other two primers selected from the UBC Primer Set #9 (University of British Columbia, Canada, available at www.github.com/btsinn/ISSRseq). Polymerase chain reaction (PCR) amplifications were carried out in a final volume of 10 µL, containing the following: 1 µL genomic DNA, 2 µL 5 × buffer (Promega, Maddison WI, USA), 0.8 µL MgCl$_2$ (25 mM, Promega), 1.6 µL dNTPs (1.25 µM, Promega), 0.5 µL bovine serum albumin (10 mg/µL, New England Biolabs, Ipswich, MA, USA), 3 µL dH$_2$O, 0.1 µL Go-*Taq* DNA polymerase (5 U/µL, Promega), and 1 µL of primer (10 µM). Thermocycling conditions were: 95 °C for 2 min (one cycle), 95 °C for 30 s, 48 °C for 30 s, 72 °C for 1 min 30 s (35 cycles), and a final extension at 72 °C for 2 min (one cycle). All PCRs contained a negative control to assess evidence for contamination. Amplified products were electrophoresed on 2% agarose gels at 100 volts for 2 min and 45 volts for 16 h and 40 min in a 4 °C cold room, and then viewed under ultraviolet light, and photographed. Sizes of amplified bands were approximated *via* comparison to a 100-bp DNA ladder. Preliminary results identified four primers (two per species) that produced informative data (Data S2), and these were selected for population-level screening.

A given allele at each ISSR locus is scored as binary presence (1) *vs.* absence (0) data, and several loci are typically co-amplified with the same primer. Accordingly, an individual's gel banding profile usually contains multiple bands of different size (*i.e.*, alleles present at different loci) and represents a multi-locus genotype. To standardize scoring of band sizes across gels and to ensure repeatability of banding profiles, PCR products from each individual were run two to three times with strategic re-ordering of individuals across gels so as to provide key side-by-side comparisons and scoring of each profile performed by two people. Only those loci and individuals that yielded reproducible results were included in downstream analyses.

## Habitat characterization

During spring and fall sampling 2015–2017, we measured stream channel characteristics. This included channel wetted width at four evenly spaced transects within our sampled reach. Using Wolman pebble counts procedures (*Wolman, 1954*; *Barnett et al., 2020*), we analyzed habitat complexity across the bankfull channel width. Ten zig-zag transects from one bank to the other were sampled at each site, with 10 points in each transect (100 sampling points/site). At each sampling point, we measured the intermediate axis of substrate. Between adjacent sampling points, we visually estimated the percentage of streambed covered by vegetation and counted number of pieces of large woody debris (LWD) (*Bain & Stevenson, 1999*).

## Diversity metrics

Species richness was measured following *Chao (1984)*, using the Chao-1 metric, which extrapolates the probability of undetected species within each site from the number of rare species detected (*i.e.*, singletons). Chao-1 species richness was calculated using the "chao1" function of the *fossil* package in R software (version 4.2.1; R project for Statistical Computing, Vienna Austria) (*R Core Team, 2022*).

To quantify cumulative multispecies abundance, we counted the number of crayfish collected after electrofishing and kick seining and used total number of individuals and area sampled to calculate the number of crayfishes collected/100 m$^2$. The cumulative multispecies abundance was summed across two sampling rounds per site.

Within-population genetic diversity was calculated using several different metrics. For mtCOI sequence data, we used DnaSP v.5.10.01 (*Librado & Rozas, 2009*) to calculate haplotypic diversity ($h$d; *Nei, 1987*). Notably, $h$d treats mtCOI haplotypes as a multi-state unordered variable. To capture information on how different haplotypes within a population were from one another, we also calculated nucleotide diversity ($\pi$; *Nei, 1987*). One of the limitations of both $h$d and $\pi$ is that they are calculated using only those haplotypes present within a given local population, and so these metrics can suffer from issues of small sample sizes. Accordingly, we also explored the utility of *Faith*'s *(1992)* Phylogenetic Diversity (PD) as means of jointly considering all of the available mtCOI sequence data for a given focal species when calculating population-specific diversity values. Briefly, PD is the sum of branch lengths on phylogenetic tree uniting all "taxa" (*i.e.*, haplotypes present within a location/population), back to the root of the tree. For both

*F. erichsonianus* and *F. validus*, rooted maximum-likelihood (ML) phylogenetic trees were estimated in MEGA v.10.2.6 (*Kumar et al., 2018*). Given that the use of an appropriate outgroup is important for PD, yet we are not aware of a published genus-level phylogeny for *Faxonius*, a phylogenetic analysis was conducted using mtCOI sequence data from 71 formally recognized species available in NCBI's nucleotide database (see Data S4). This identified *F. spinosus* as the sister taxon of *F. erichsonianus*, whereas a clade including both *F. cooperi* and *F. pagei* was sister to *F. validus*. For each of the two focal species and their associated outgroup(s), the best-fit model of nucleotide evolution was determined *via* Akaike information criterion (AIC) model selection, and an ML tree was estimated using the following search settings: missing data = partial deletion (cut-off: 95%), maximum parsimony starting tree, nearest-neighbor-interchange branch swapping, and branch swap filter = moderate. Node support was assessed *via* 500 bootstrap replicates. The resulting tree was exported in Newick format containing tree topology plus estimated branch lengths, and PD was then calculated using the *picante* package (v1.8.2; *Kembel et al., 2010*) in R.

For the ISSR data, within-population genetic diversity was calculated as the proportion of polymorphic loci (PPL) (*i.e.*, number of loci that were polymorphic among individuals at local site, divided by total number of loci screened for the focal species). Because there was no correlation between PPL and number of individuals within a population sample (Pearson correlation: *F. validus*: $r = 0.27$, $P = 0.22$; *F. erichsonianus*: $r = 0.29$, $P = 0.16$), subsequent rarefaction correction of PPL was not applied (Table 2).

## Species richness, abundance, and genetic diversity correlations

For the focal *Faxonius* species at all study sites, collectively, we investigated SGDC assessing the relationships between species richness and each of the four genetic diversity metrics separately (*i.e.*, $h$d, $\pi$, and PD for mtCOI sequences, and PPL for ISSR markers). To do this, we calculated the Pearson correlation coefficient and asymptotic confidence intervals based on Fisher's Z transformation using the "cor.test" function of the *stats* package in R to determine significance (*R Core Team, 2022*). At each of the same 32 sampling sites for *F. validus* and *F. erichsonianus* (Fig. 1), we tested the MIH by assessing the correlation between species richness and cumulative multispecies abundance of all crayfishes, again using Pearson correlation coefficient and confidence intervals to determine significance, calculated in R. AGDC was assessed in the same way as SGDC, except that species richness was replaced by cumulative multispecies abundance.

## Relationships between habitat characteristics, community- level diversity, and population-level diversity

To clarify the extent to which stream fragmentation and environmental characteristics (size, connectivity, and habitat complexity) may impact SGDC, MIH and/or AGDC, we examined whether species richness, abundance, and genetic diversity metrics showed any relationships with stream channel characteristics (see *Habitat characterization*, above) using linear models. If stream channel characteristics affect the two levels of diversity in a different way (*e.g.*, positive relationship between stream size and genetic diversity *vs.*

negative relationship between stream size and species richness), we would expect this to lead to no or negative correlation between different diversity metrics, thus resulting in no support for SGDC, MIH and/or AGDC.

To assess impacts of stream channel characteristics on diversity correlations, we calculated the median wetted width, percent vegetation, substrate size (*i.e.*, D50), and LWD from spring and fall sampling for each site. All sites within unimpounded streams were characterized as connected, and all sites within impounded streams were characterized as fragmented. Because dams and associated impoundments can have far-reaching effects up- and downstream of impoundments (*Falke & Gido, 2006*; *Johnson, Olden & Zanden, 2008*), all sites sampled within impounded streams have the potential to be fragmented (*e.g.*, isolation of upstream sites, alterations of seasonal flow patterns downstream and habitat modification both up- and downstream) (*Yeager, 1993*). We fit linear models with least squares estimates using the 'lm' function in with *stats* package in R. In these models, stream characteristics were treated as the independent variables (potential predictors), whereas the different metrics for species richness, cumulative multispecies abundance, or genetic diversity were treated as the dependent variable (response). We included 2-way interactions of stream characteristics and binary stream type classification (*i.e.*, connected *vs.* fragmented). We used the *MuMIn* R package (*Barton & Anderson, 2002*) to analyze all possible models. Model selection was based on corrected Akaike information criterion (AICc) because sample sizes were small relative to the number of estimated parameters (*Burnham & Anderson, 2004*). We compared alternative models by weighting their level of data support (*Hurvich & Tsai, 1989*), with delta AICc values ≤2 representing the best-supported models. We calculated relative variable importance (RVI; number of models predictor variable appears in/number of total models) scores for each predictor variable, based on variables appearance in the AICc-best models. Predictors with RVI >0.5 were considered most important. If there were significant stream characteristics by stream type (connected or fragmented) interactions, pairwise comparisons between each stream type and stream characteristic were done. Tukey HSD *P*-value adjustment approach (*Sokal & Rohlf, 1981*) was used to correct for the effect of multiple comparison on the family-wise error rate.

## RESULTS

### Crayfish collections, and characterization of genetic diversity

Across all sites, we collected 12 crayfish species, with six and eight species collected in the Bear Creek and Cahaba River drainages, respectively (Table 3). Additionally, nine crayfish species were collected in both impounded and unimpounded streams. Cumulative multispecies abundance of individuals varied greatly between sites ($N$ crayfish/100 m$^2$ = 0.001–0.282), with an average of 0.032 crayfish collected per 100 m$^2$ (Datas S5 and S6). The highest densities of crayfishes were collected in Rock Creek (0.723) and lowest in Shades Creek (0.016) (Table 3).

For *F. validus*, we successfully sequenced 143 individuals, obtaining a 618-bp mtCOI alignment, with 25 polymorphic sites and 28 unique haplotypes (Table 2; Data S3) (data from *Barnett et al., 2020* assessed; GenBank accession numbers MN053979–MN054006,

**Table 3 Cumulative multispecies abundance of crayfish (measured as a density: crayfish individuals/100 m$^2$) in upstream (Up) and downstream (Dn) sections of impounded and unimpounded streams in the Bear Creek and Cahaba River drainages, Alabama.**

| Drainage | Crayfish | Impounded | | | | Unimpounded | | Total |
|---|---|---|---|---|---|---|---|---|
| Bear creek | | Cedar-up | Cedar-dn | Little bear-up | Little bear-dn | Rock-up | Rock-dn | |
| | *Faxonius validus* | 0.0392 | 0.0246 | 0.0768 | 0.0624 | 0.5223 | 0.0162 | **0.7416** |
| | *Faxonius erichsonianus* | 0.0367 | 0.0163 | 0.0498 | 0.0205 | 0.0346 | 0.0440 | **0.2019** |
| | *Cambarus striatus* | 0.0002 | 0.0000 | 0.0018 | 0.0003 | 0.0731 | 0.0077 | **0.0832** |
| | *Faxonius compressus* | 0.0000 | 0.0000 | 0.0000 | 0.0024 | 0.0000 | 0.0239 | **0.0263** |
| | *Faxonius etnieri* | 0.0000 | 0.0000 | 0.0000 | 0.0000 | 0.0000 | 0.0031 | **0.0031** |
| | *Lacunicambarus dalyae* | 0.0000 | 0.0000 | 0.0004 | 0.0004 | 0.0000 | 0.0023 | **0.0031** |
| | Total | **0.0762** | **0.0409** | **0.1288** | **0.0860** | **0.6300** | **0.0972** | **1.0591** |
| Cahaba River | | Little cahaba-up | Little cahaba-dn | | | Shades-up | Shades-dn | |
| | *Faxonius virilis* | 0.0153 | 0.0048 | | | 0.0034 | 0.0028 | **0.0263** |
| | *Faxonius erichsonianus* | 0.0030 | 0.0025 | | | 0.0019 | 0.0058 | **0.0133** |
| | *Cambarus coosae* | 0.0001 | 0.0010 | | | 0.0000 | 0.0000 | **0.0011** |
| | *Procambarus clarkii* | 0.0007 | 0.0000 | | | 0.0000 | 0.0004 | **0.0011** |
| | *Cambarus striatus* | 0.0002 | 0.0003 | | | 0.0000 | 0.0021 | **0.0026** |
| | *Procambarus acutus* | 0.0005 | 0.0000 | | | 0.0000 | 0.0000 | **0.0005** |
| | *Faxonius spinosus* | 0.0001 | 0.0000 | | | 0.0000 | 0.0000 | **0.0001** |
| | *Cambarus acanthura* | 0.0000 | 0.0000 | | | 0.0000 | 0.0001 | **0.0001** |
| | Total | **0.0200** | **0.0086** | | | **0.0053** | **0.0111** | **0.0450** |

Data S3). For *F. erichsonianus*, we obtained a 640-bp mtCOI alignment, with 68 polymorphic sites and 42 haplotypes from 173 individuals (data from *Barnett et al., 2020* assessed; GenBank accession numbers MN054007–MN054048, Data S3). We obtained ISSR data from 109 *F. validus* and 95 *F. erichsonianus* individuals. *Faxonius validus* and *F. erichsonianus* included in ISSR analyses were a subset of those used in mtCOI assessments. We assessed a minimum of three individuals per site (mean four individuals/site) and only used individuals that yielded reproducible results. *Nelson & Anderson (2013)* showed that genetic diversity estimates were similar when using five compared to 10 individuals per site, suggesting that our sample sizes are likely reasonable. ISSR primers yielded 24 and 34 polymorphic loci for *F. validus* and *F. erichsonianus*, respectively (Datas S5 and S6). While this is a relatively low number of polymorphic loci (*Nelson & Anderson, 2013*), studies have shown that using 20–30 dominant markers can yield acceptable results for population genetic assessments (*Vandergast et al., 2009*; *Guasmi et al., 2012*; *Nelson & Anderson, 2013*).

## Habitat characterizations

Crayfish were collected in medium (median wetted width = 11.81 m) sized streams with mostly pebble substrate (median substrate size = 25 mm) and a relatively wide range of LWD (2–25 pieces of LWD; median = 10.0) and percent vegetation (6–32%; median = 15.6) (Table 4; Datas S5 and S6).

**Table 4 Fragmentation status and median values for stream channel parameters (range) from crayfish surveys.** D50, median substrate size; LWD, number of pieces of large woody debris.

| Fragment status | Little Bear Fragmented | Cedar Fragmented | Rock Connected | Little Cahaba Fragmented | Shades Connected |
|---|---|---|---|---|---|
| Wetted Width (m) | 11.0 (5.1–13.2) | 13.9 (9.1–18.4) | 8.2 (3.1–11.7) | 13.4 (7.5–17.8) | 12.5 (10.5–13.3) |
| D50 (mm) | 24.8 (16–599) | 26.8 (6–1,013) | 19.1 (12–1,039) | 25.4 (17–79) | 17.5 (2–25) |
| Aquatic vegetation (%) | 14.5 (3–29) | 16.3 (9–29) | 27.8 (17–34) | 8.2 (7–11) | 11.2 (11–19) |
| LWD | 9.0 (2–25) | 8.0 (3–16) | 6.0 (2–13) | 10.8 (9–14) | 16.0 (13–20) |

**Table 5 Correlations (Pearson $r$) and confidence intervals between species diversity-abundance (*i.e.*, more individuals hypothesis, MIH), species-genetic diversity (SGDC), and abundance-genetic diversity (AGDC) for *Faxonius validus* and F. *erichsonianus*.** Genetic diversity was measured using mitochondrial COI sequence data (nucleotide diversity ($\pi$), haplotypic diversity ($h$d), and phylogenetic diversity (PD)), and ISSR nuclear marker data (proportion of polymorphic loci (PPL)). Significant relationships are shown in bold.

| | Pearsons $r$ | Confidence interval | $P$ value |
|---|---|---|---|
| (A) *Faxonius validus* | | | |
| SGDC— $\pi$ | 0.01 | [−0.393 to 0.413] | 0.96 |
| $h$d | −0.06 | [−0.450 to 0.354] | 0.79 |
| PD | 0.08 | [−0.306 to 0.437] | 0.70 |
| PPL | 0.02 | [-0.402 to 0.441] | 0.92 |
| AGDC— $\pi$ | −0.36 | [−0.666 to 0.052] | 0.09 |
| $h$d | −0.45 | [−0.723 to −0.059] | **0.03** |
| PD | 0.04 | [−0.340 to 0.405] | 0.85 |
| PPL | 0.32 | [−0.121 to 0.651] | 0.15 |
| (B) *Faxonius erichsonianus* | | | |
| SGDC— $\pi$ | 0.15 | [−0.239 to 0.493] | 0.46 |
| $h$d | −0.06 | [−0.421 to 0.323] | 0.77 |
| PD | −0.12 | [−0.489 to 0.292] | 0.58 |
| PPL | −0.12 | [−0.489 to 0.292] | 0.68 |
| AGDC— $\pi$ | 0.08 | [−0.115 to 0.584] | 0.17 |
| $h$d | −0.10 | [−0.280 to 0.459] | 0.60 |
| PD | 0.04 | [−0.340 to 0.405] | 0.85 |
| PPL | 0.07 | [−0.336 to 0.451] | 0.75 |
| (C) MIH | −0.16 | [−0.527 to 0.262] | 0.46 |

## Species richness and genetic diversity estimates

Regarding community-level diversity, Chao-1 species richness ranged from one to seven species, with an average of four species per site (Datas S5 and S6). Regarding population-level diversity, *F. validus* mtCOI haplotypic diversity and nucleotide diversity were typically lower than that of *F. erichsonianus* (mean = 0.57 [SD = 0.20] *vs.* 0.63 [0.17]; mean = 0.002 [0.001] *vs.* 0.003 [0.002], respectively Table 2). Likewise, phylogenetic diversity (PD) values, measured in branch length units of substitutions per site, were

**Table 6 Linear model results of the relationship between crayfish diversity metrics (species richness, abundance, population genetic diversity) and stream characteristics.** Results include variables from the models that were within two $AIC_c$ units of the best model. Stream characteristics are listed by decreasing relative variable importance (RVI). Null model indicates that the null model was the best model. Pairwise results for variables with significant interactions are shown in Fig. 2. N, number of models within two $AIC_c$ units of the best model. SE, standard error. RVI, relative variable importance (parameters with RVI of 1.00 were included in all of the best models). D50, median substrate size (mm). LWD, large woody debris (number of pieces). $\pi$, nucleotide diversity. hd, haplotypic diversity. PPL, proportion of polymorphic loci. *Indicates P values $\leq 0.05$. **Indicates P values $\leq 0.01$. − Indicates that no parameters were assessed because null model was the best model.

| Model | $R^2$ | N | Estimate | SE | RVI |
|---|---|---|---|---|---|
| Chao 1 species richness | 0.36 | 1 | | | |
| Stream type × stream width** | | | 0.417 | 0.310 | 1.00 |
| Cumulative multispecies abundance | 0.97 | 2 | | | |
| Stream type × D50** | | | <−0.001 | <0.001 | 1.00 |
| Stream type × LWD** | | | −0.001 | 0.001 | 1.00 |
| Stream width** | | | −0.001 | <0.001 | 0.50 |
| *Faxonius validus* $\pi$ | 0.19 | 1 | | | |
| Stream width* | | | <0.001 | <0.001 | 1.00 |
| *Faxonius validus* hd | 0.22 | 1 | | | |
| Stream width* | | | 0.032 | 0.012 | 1.00 |
| *Faxonius validus* PPL | – | – | | | |
| Null model | | | – | – | – |
| *Faxonius erichsonianus* $\pi$ | – | – | | | |
| Null model | | | – | – | – |
| *Faxonius erichsonianus* hd | – | – | | | |
| Null model | | | – | – | – |
| *Faxonius erichsonianus* PPL | – | – | | | |
| Null model | | | – | – | – |

generally lower for *F. validus* than *F. erichsonianus* (mean = 0.006 [0.004] *vs.* 0.015 [0.014], respectively; Table 2). Conversely, the ISSR-based proportion of polymorphic loci (PPL) showed close equivalence between the two focal species (mean PPL = 0.87 [0.11] *vs.* 0.85 [0.07] for *F. validus* and *F. erichsonianus*, respectively; Table 2).

## Species richness, abundance, and genetic diversity correlations

Pearson correlation tests showed no significant correlations between species richness and genetic diversity for any of the mtCOI- or ISSR-based diversity metrics for *F. validus* or for *F. erichsonianus* (Tables 5A, 5B). Additionally, Pearson correlation tests showed no significant correlations between cumulative multispecies crayfish abundance and species richness (P = 0.46; Table 5C). As such, neither SGDC nor MIH were supported.

An AGDC was evident in one focal species (Table 5A). For *F. validus*, Pearson correlation tests showed a negative relationship between cumulative multispecies abundance and genetic diversity for two of the three mtCOI-based metrics (hd: r = −0.45,

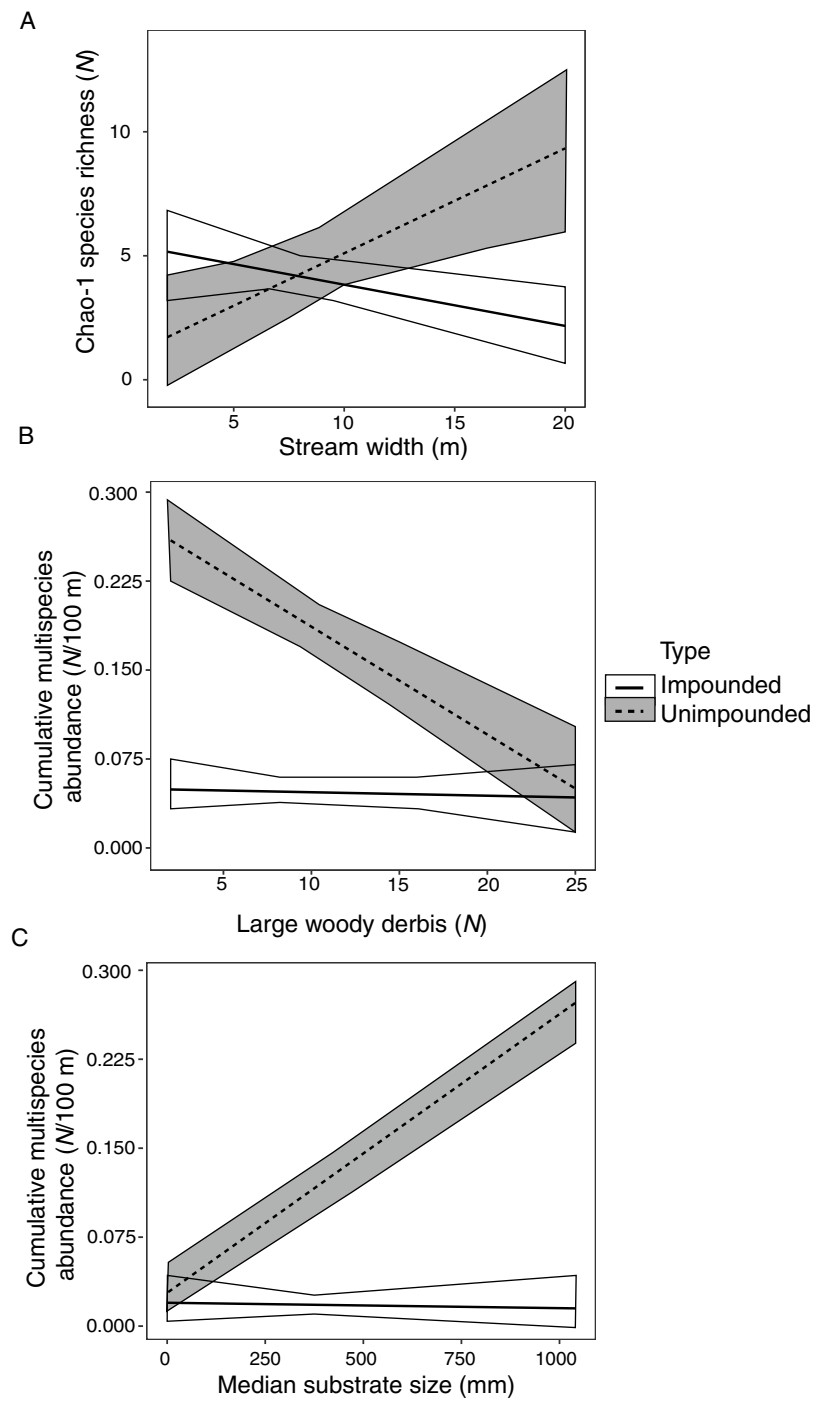

**Figure 2 Impounded and unimpounded stream comparisons of mean (standard error) Chao-1 species richness (A) and cumulative multispecies abundance (B, C) among mean stream width (A), number of pieces of large woody debris (B), and median substrate size (C).** Only relationships with significant interactions in linear models are displayed.

$P = 0.03$; π: $r = -0.36$, $P = 0.09$), although only the abundance-$h$d correlation was significant at the 0.05-level. However, the Pearson correlation test showed no significant correlation between abundance and ISSR-based PPL (Table 5A). For *F. erichsonianus*,

Pearson correlation tests showed no significant relationship between cumulative multispecies abundance and genetic diversity for any mtCOI ($h$d, $\pi$, PD) or ISSR (PPL) diversity measure ($r = -0.10–0.27$, $P > 0.05$; $r = 0.07$, $P = 0.75$, respectively) (Table 5B).

## Association between environmental characteristics, community-level diversity, and population-level diversity

Species richness was significantly correlated with stream width, explaining 36% of species richness variation (Table 6). However, this correlation varied depending on stream type (fragmented *vs.* connected) (Fig. 2A). There was a positive relationship between species richness and stream width in unimpounded streams, but a negative relationship in impounded streams.

Cumulative multispecies abundance was significantly correlated with stream width, LWD, and substrate size, which explained 97% of crayfish abundance variation (Table 6). Cumulative multispecies abundance increased with decreasing stream width, and its relationship with LWD and substrate size varied depending on stream type (Figs. 2B, 2C). There was a significant negative relationship between cumulative multispecies abundance and amount of LWD in unimpounded streams, but no relationship with LWD in impounded streams (Fig. 2B). Additionally, there was a significant positive relationship between cumulative multispecies abundance and substrate size in unimpounded streams, but no relationship in impounded streams (Fig. 2C).

Stream width was significantly positively correlated with *F. validus* nucleotide and haplotype diversity, explaining 19% and 22% of their variation, respectively (Table 6). There was no significant relationship between ISSR-based PPL and stream characteristics. Additionally, there were no significant relationships between any *F. erichsonianus* genetic diversity metrics and stream characteristics (Table 6).

## DISCUSSION

The most salient findings of this study were that no positive SGDCs, MIH, or AGDCs were detected. These findings have several important implications for the conservation of crayfish diversity. Given no positive correlations, separate strategies for conserving species richness, abundance, and genetic diversity seem appropriate. Here, we showed that fragmentation changed the relationship between environmental factors and community-level diversity metrics, with species richness and multispecies abundance consistently lower in fragmented habitats. *Barnett et al. (2023)* showed that in less complex habitats (*e.g.*, aquatic vegetation and woody debris), more predatory fishes, higher minimum temperatures, and less variable discharges led to lower densities and diversity of crayfishes in impounded *versus* unimpounded streams. Thus, conservation practices restoring habitat complexity, mimicking natural flow regimes, and increasing connectivity in fragmented riverine systems may increase community-level diversity in these systems. Additionally, riverine systems where community-level diversity is high may still deserve conservation priority to prevent the loss of genetic diversity, as this may nonetheless be

low. Efforts to increase dispersal and gene flow between populations (*e.g.*, barrier removal, habitat restoration, hydrologic restoration) may be needed to increase genetic diversity.

Even though no positive correlations between diversity metrics were detected, we did detect a negative correlation between cumulative multispecies abundance and mtCOI-based genetic diversity ($h$d, and π) for *F. validus*, although only $h$d was statistically significant. Negative AGDCs indicate that conditions favoring high abundance of crayfishes were coupled with low genetic diversity. However, studies have suggested that large populations maintain high genetic diversity (*Frankham, 2010*; *Allendorf, Luikart & Aitken, 2013*). Nonetheless, in river ecosystems, studies report a downstream increase in genetic diversity due to increased downstream dispersal with waterflow (*Ritland, 1989*; *Kikuchi, Suzuki & Sashimura, 2009*; *Alp et al., 2012*; *Paz-Vinas et al., 2015*), while crayfish abundance increases upstream due to positive relationships with hydrologic variability of headwater streams (*Flinders & Magoulick, 2003*; *Yarra & Magoulick, 2018*). Similarly in this study, crayfish abundance was highest in upstream sites in both impounded and unimpounded streams in the Bear Creek drainage, which could be due to crayfish burrowing capabilities, along with reduced predation risk in upstream sites (*Flinders & Magoulick, 2003*; *Yarra & Magoulick, 2018*, *2020*). Conversely, genetic diversity was highest at downstream sites which could be due to higher dispersal from tributaries entering the stream and passive movement of crayfish downstream during high flow events (*Maude & Williams, 1983*). *Barnett et al. (2020)* also showed that higher gene flow occurred from up- to downstream than down- to upstream among most *F. validus* populations in Bear Creek drainage streams, with impoundments negatively impacting upstream gene flow. Negative AGDC trends were not evident in *F. erichsonianus* (Table 5B). This difference between species may be due to contrasting habitat preferences, with *F. erichsonianus* commonly collected in small to large streams under rocks and in leaf litter (*Bouchard, 1972*; *Hobbs, 1981*), whereas *F. validus* is found only along the margins of small to medium sized streams and in temporary streams that dry seasonally (*Bouchard, 1972*; *Cooper & Hobbs, 1980*). Sampling within watersheds revealed that *F. validus* dominates at sites furthest upstream, whereas *F. erichsonianus* dominates at sites near impoundments and midpoints within unimpounded streams, as well as in our furthest downstream sites (*Barnett et al., 2020*, *2022*). Additionally, *F. erichsonianus* were collected in the Cahaba River drainage during spring 2016 and fall 2017, while both species were collected in the Bear Creek drainage during spring and fall of 2015. This sampling scheme could potentially introduce a geographical and temporal bias to the study. However, in previous studies by *Barnett et al. (2022*, *2023)* which assessed crayfish community structure in Bear Creek drainage streams between 2015–2017, community structure differences were not detected between years indicating that crayfish communities may not have seen great changes within this timeframe. Furthermore, positive correlations were detected between stream width and *F. validus* haplotype and nucleotide diversity, whereas no correlations were detected between stream width and *F. erichsonianus* genetic diversity metrics. Species habitat preferences and trend differences between stream size and genetic diversity may explain species-specific differences in support for AGDCs.

Negative AGDC trends were not evident with our nuclear DNA (nDNA) assessments. Discrepancies in AGDC trends between ISSR and mtCOI markers may reflect differences in effective population sizes and mutation rates among genes, and/or sex-biased dispersal. Nuclear DNA has roughly four times the $N_e$ of mitochondrial DNA (mtDNA) (assuming an equal sex ratio among breeding adults). The smaller $N_e$ of mtDNA potentially allows it to capture the signal of demographic events that may not leave a mark in nDNA loci (*Vandergast et al., 2009*; *Eytan & Hellberg, 2010*), which may explain why we only detected a negative relationship with mtCOI markers. Nuclear DNA also captures biparental inheritance and thus dispersal of both males and females, while mtDNA is maternally inherited and therefore only captures information on dispersal of females. Furthermore, sex-biased movement could potentially explain differences between our mtDNA and nDNA findings, however, there is currently no evidence of sex-biased dispersal within other crayfish (*Gherardi, Tricarico & Ilhèu, 2002*; *Bubb, Thom & Lucas, 2004*; *Wutz & Geist, 2013*; *Galib et al., 2022*).

Like other freshwater macroinvertebrate assessments, no SGDCs were detected in our study (*Seymour et al., 2016*; *Watanabe & Monaghan, 2017*; *Petersen et al., 2022*). Previous studies that found positive SGDCs indicate that environmental and physical variation significantly correlated with species richness (*He et al., 2008*; *Lamy et al., 2013*), suggesting that species richness may be locally selected, which then influences genetic diversity. In our study, we found that stream width was correlated with species richness, but this correlation had opposite trends in impounded (positive correlation) and unimpounded (negative correlation) streams (Fig. 2). Stream width was also correlated with *F. validus* mtCOI population genetic diversity metrics. Unlike species richness, *F. validus* population genetic diversity was positively correlated with stream width no matter the stream type. Thus, fragmentation may be impacting only species richness, decoupling SGDCs. Additionally, no tested environmental factor was correlated to *F. erichsonianus* population genetic diversity, indicating stream width is not a driver for all crayfishes within this system.

Positive species-abundance correlations (MIH) are mainly expected in communities where interspecific competition is relatively low and environmental factors (*e.g.*, habitat heterogeneity, land use intensity) impact most species similarly (*Vellend & Geber, 2005*; *Storch, Bohdalková & Okie, 2018*). Crayfish are not all impacted the same by environmental factors (*Adams, 2013*; *Mouser, Mollenhauer & Brewer, 2019*; *Barnett et al., 2020*, *2023*), and there is high interspecific competition between co-occurring species (*Blank & Figler, 1996*; *Mouser, Mollenhauer & Brewer, 2019*). For example, in the Ozark Highlands ecoregion of Missouri, the presence of crayfishes that were strong competitors resulted in lower occurrence of species that were not strong competitors (*Mouser, Mollenhauer & Brewer, 2019*). Additionally, abundance of some crayfish species increased in Alabama streams with little habitat heterogeneity, while others were found only in sites with high habitat heterogeneity (*Barnett et al., 2022*). Crayfish also have different burrowing capabilities (*Hobbs, 1981*), which may lead to contrasting responses to fragmentation and habitat heterogeneity. Indeed, species that we sampled in local communities ranged from tertiary to secondary burrowers. In the present study, species richness and cumulative multispecies abundance were correlated with different stream

environmental characteristics. As such, changes in stream characteristics could impact one diversity metric but not the other.

Many SGDC, MIH, and AGDC studies have reported contrasting results, depending on the focal species (*Scribner et al., 2001*; *Wei & Jiang, 2012*; *Watanabe & Monaghan, 2017*; *Storch, Bohdalková & Okie, 2018*) and the environmental context. Assuming ecological similarity, focal species that are common are expected to show positive SGDCs, while rare species are more likely to differ from the overall community, with population sizes and genetic diversity of rare species often not positively correlated with locality area and thus also not positively correlated with abundance and richness of the overall community (*Velend, 2005*). In this study, we selected the most abundant species collected in our study systems (making up ≥30% of individuals collected) as focal species for genetic diversity assessments. *Faxonius erichsonianus* is also relatively abundant throughout the southeastern region, occurring in six southeastern states from western Tennessee south to northern Mississippi and northwestern Georgia, north to Virginia (*Hobbs, 1981*). Conversely, *Faxonius validus* occurs only in the Tennessee and Black Warrior River basins in northern Alabama and southern Tennessee (*Cooper & Hobbs, 1980*; *Hobbs, 1989*). Unlike these two focal species, other species within the study streams that make up the local community are more broadly distributed throughout the eastern US (*e.g.*, *Procambarus acutus*) or the entire US (*e.g.*, *F. virilis* and *P. clarkii*), and are invasive in some environments (*e.g.*, *F. virilis*, *P. acutus*, and *P. clarkii*). Thus, the geographic range and commonality of our focal species is much less than other species in our study systems, indicating dispersal and niche limitations in our focal species (*Astorga et al., 2012*). Additionally, our sampling sites were in the Eastern Highland region, which has a pre-Pleistocene origin and is likely the center of origin of *Faxonius* (*Crandall, Templeton & Neigel, 1999*). Focal species responses to glaciation and sea-level fluctuation along with dispersal differences between species and sexes could drive differences detected between *F. validus* and *F. erichsonianus*, as well as ISSR (bi-parentally inherited) and mtCOI (maternally inherited) markers (*Crandall, Templeton & Neigel, 1999*; *Mayden, 1987*). Furthermore, specific demographic histories of population bottlenecks and expansions are unknown for these species. Moreover, ecological, evolutionary, and demographic differences between our focal species and other members of the overall community may have contributed to the absence of significant correlations between diversity measures.

The effects of habitat fragmentation and modification, such as those caused by impoundments, have long been recognized as a major threat to biodiversity (*Vandergast et al., 2007*; *Bessert & Ortí, 2008*; *Quadroni et al., 2016*), with life history characteristics such as dispersal ability and physiological tolerances often determining the degree of impact (*Luoy et al., 2007*; *Reid et al., 2008*; *Alp et al., 2012*). Both community-level diversity correlations with stream characteristics differed between impounded and unimpounded streams. Conversely, genetic diversity correlations with stream characteristics did not differ between impounded and unimpounded streams. These findings indicate different responses of community-level diversity and population-level genetic diversity to environmental conditions. Nonetheless, we could not assess differences between diversity correlations from impounded and unimpounded streams separately because only nine of

the 32 sampling sites were in unimpounded streams, which provides low power for detecting any differences.

This study used both nuclear (ISSR) and mitochondrial (mtCOI) markers to assess within-population genetic diversity. While methods such as next-generation sequencing are becoming increasingly common for genetic diversity assessments, in conservation planning the need remains for simple, cost-effective, yet robust methods. Numerous studies highlight the reliability, simplicity, and cost effectiveness of ISSR markers when assessing genetic variation (*Grativol et al., 2011*; *Sarwat, 2012*; *Saha et al., 2020*). The mtCOI gene is the most commonly used genetic marker for crayfish assessments (*Fetzner & DiStefano, 2008*; *Barnett et al., 2020*; *Cabe et al., 2022*; *Lovrenčić et al., 2022*). Therefore, it may be of broad interest to understand if the diversity metrics estimated from this marker correlate with community-level metrics, so that managers can understand the potential for re-purposing existing mtCOI datasets. Additionally, using both ISSR and mtCOI markers provides replicate samples of the demographic history of focal species (*Brito & Edwards, 2009*). However, markers may estimate demographic history differently due to mechanisms affecting their evolution, $N_e$, or rates of recombination (*Graur & Li, 2000*; *Hare, 2001*; *Brito & Edwards, 2009*; *Eytan & Hellberg, 2010*). For example, ISSRs are transmitted biparentally, may have interlocus recombination and a large $N_e$. Conversely, mtCOI is transmitted maternally as a single nonrecombining block and has a comparatively small $N_e$. This small $N_e$ gives mtDNA the ability to detect more recent changes to a population than nDNA (*Moore, 1995*), while nDNA has the ability to provide replicate samples of the underlying demographic history affecting the genome of an organisms and coalescent process (*Carling & Brumfield, 2007*). Thus, these markers should complement each other (*Eytan & Hellberg, 2010*; *Garrick, Caccone & Sunnucks, 2010*). Nonetheless, one shortcoming of our study is the relatively small number of ISSR loci assessed (24 and 34 polymorphic loci for *F. validus* and *F. erichsonianus*, respectively). While similar numbers of loci have been shown to be reliable in other studies (*Vandergast et al., 2009*; *Guasmi et al., 2012*; *Nelson & Anderson, 2013*), the minimum number of loci required to yield acceptable results depends on the analyses being performed and level of genetic differentiation among populations (*Nelson & Anderson, 2013*). Thus, future studies should add more nDNA loci to assess correlations between diversity metrics. Additionally, the differences between the suite of genetic diversity metrics used in this study indicates that other types of nDNA markers should also be assessed.

## CONCLUSIONS

We assessed evidence for species-genetic diversity correlations (SGDCs), more individuals hypothesis (MIH), and multispecies abundance-genetic diversity correlations (AGDCs) within crayfish communities in impounded and unimpounded streams in the southeastern US. Our results indicated a significant relationship between cumulative multispecies abundance and genetic diversity (AGDC) for one of the focal species, but unexpectedly, this AGDC was negative. Notably, the level of support for this negative AGDC differed across genetic marker types, and even among different metrics for mtCOI variation. We also investigated the association of several environmental factors with species richness,

population genetic diversity, and cumulative multispecies abundance. In this context, we found that fragmentation status affected the relationship between several environmental factors and species richness, population genetic diversity, and cumulative multispecies abundance, which could explain why there was generally little or no support for SGDC, MIH and/or AGDCs.

Crayfish are among the most threatened North American taxa, and the need for crayfish conservation is particularly urgent (*Taylor et al., 2019*). However, most conservation planning is focused at the community-level, with less emphasis on population-level genetic diversity. Our study showed that community-level diversity was not positively correlated with population-level genetic diversity, and eco-evolutionary processes influencing genetic diversity were not the same as those influencing community-level diversity. Thus, conservation at the community-level may not be protecting population diversity and could potentially lead to a loss of population-level diversity with detrimental consequences for the species in the long term. Accordingly, managers need to survey both community- and population-level diversity, as well as habitat diversity and integrity and set separate conservation actions for each hierarchical level of biodiversity (*i.e.*, decreasing sedimentation may increase multispecies abundance). Additionally, efforts to preserve evolutionary and ecological processes is crucial for the long-term conservation of species, particularly in the face of habitat alteration/fragmentation and environmental change. Future studies assessing crayfish species across a larger geographic range in fragmented and connected habitats will give further insight on how diversity metric correlations, ecological preferences, and interspecific interactions impact crayfish communities on a broader scale and in different riverine ecosystems. Understanding the relationship between biodiversity levels for vulnerable taxonomic groups will not only give insight to factors impacting at-risk crayfishes, it will allow conservationists to protect the numerous ecosystem services (*e.g.*, transferring energy to higher level organisms, creating habitat for other organisms through burrow creation) provided by crayfishes, with an overall protection of the existing biodiversity within these communities.

## ACKNOWLEDGEMENTS

We thank the following individuals for assistance with field collections: G. McWhirter, M. Bland, C. Smith, K. Sterling, and Z. Choice (USFS); S. McGregor, R. Bearden, Sandi Stanley, and P. Nenstiel (Geological Survey of Alabama); S. McKinney (Bear Creek Development Authority); C. Johnson (Alabama Department of Environmental Management); D. Butler (Cahaba Riverkeeper); C. Mangum (Weyerhaeuser); J. Sackreiter, J. Payne, J. Banusiewicz, L. Eveland, E. Liles, K. Forbes, and R. Smith (University of Mississippi); C. Quinn and F. Murphy (USFS contractors); B. Simms (American Fisheries Society Hutton Scholar); and E. Choice, K. Abdo, T. Reed, and S. Barnett (volunteers). We thank S. Santiago (USFS) for scoring gels. We also thank C. Sabatia for statistical review.

### Funding
Funding was provided by the USDA Forest Service Southern Research Station, University of Mississippi, and Birmingham Audubon Society. The funders had no role in study design, data collection and analysis, decision to publish, or preparation of the manuscript.

### Grant Disclosures
The following grant information was disclosed by the authors:
USDA Forest Service Southern Research Station.
University of Mississippi, and Birmingham Audubon Society.

### Competing Interests
The authors declare that they have no competing interests.

### Author Contributions
- Zanethia C. Barnett conceived and designed the experiments, performed the experiments, analyzed the data, prepared figures and/or tables, authored or reviewed drafts of the article, and approved the final draft.
- Ryan C. Garrick conceived and designed the experiments, analyzed the data, authored or reviewed drafts of the article, and approved the final draft.

### Field Study Permissions
The following information was supplied relating to field study approvals (*i.e.*, approving body and any reference numbers):
Field collections were approved by the state of Alabama. (Alabama Conservation License #s 2016064289868680 and 2017092711268680).

### Data Availability
All raw data, including diversity and stream habitat measurements are in the Supplemental File.

### Supplemental Information
Supplemental information for this article can be found online at http://dx.doi.org/10.7717/peerj.18006#supplemental-information.

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
