# Peer review of "Relationships between crayfish population genetic diversity, species richness, and abundance within impounded and unimpounded streams in Alabama, USA"

_PeerJ, doi:10.7717/peerj.18006_

## Round 0.1 · original submission · Major Revisions

Three colleagues have reviewed your manuscript and recognised the research work you have carried out. On the other hand, they pointed out problems that make the manuscript unacceptable in its current form. The most important aspect is the use of ISSRs due to the small number of markers considered and the number of individuals analysed per site. As suggested by reviewer 2, the work would still be relevant if you focussed only on COI. If you decide to leave the ISSRs in, you should at least discuss the robustness and relevance of your results. Reviewer 2 also gives very specific advice on this. Of course, you should also consider the other very detailed recommendations of the three reviewers for the revision.
I hope that the reviewers' constructive comments will enable you to revise the manuscript substantially, which is a prerequisite for its acceptance.

·

Basic reporting

1. The manuscript is overall well-written concerning language and style, but not always scientifically accurate: the use of scientific terminology must be improved in several points (see Additional comments).

2. Introduction and background are in general appropriate; but see Additional comments for minor suggestions. Literature citations are appropriate.

3. The article has a rigorous structure and follows the PeerJ format. However, not sure whether in-text citations follow PeerJ style. For example, add comma between author names and year; multiple citations should be separated by semicolon; "&" instead of "and". Please check author guidelines.

4. Figures and tables are relevant and well described. However, for the sake of clarity, perhaps a figure with a brief synopsis of your study and the tested hypotheses might help. The discussed hypothesis framework is complex and could be not easy to follow for a reader who is not familiar with SGDC studies. See for example Fig. 1 in: Bucholz et al., 2023. Community‐wide correlations between species richness, abundance and population genomic diversity in a freshwater biodiversity hotspot. Molecular Ecology. However, please take this only as a suggestion.

5. All the raw data have been provided, except perhaps for raw dataset for ISSR markers (see Additional comments).

Experimental design

1. Original research is within the scope of the journal.

2. Research questions are well defined and addressed in detail; the tested hypothesis framework is
complex and carefully designed, with careful considerations of all the potential underlying factors.

3. Field and lab works are considerable; all the analyses were performed rigorously.

4. In general, methods are well described and reproducibility of analyses is granted. However, in Methods (line 209) and Results (line 389), the authors cited Barnett et al. 2022 when speaking about samples collection and when reporting genetic results. I don't understand if the authors relied on this previous study, for these data. Personally, I would find it acceptable, particularly considering that those data are here analyzed together with other kind of information and within a complex hypothesis testing framework. However, I think that this must be more explicitly stated (or clarified).

Validity of the findings

1. This study could represent a valuable contribution in the context of species-genetic diversity correlation (SGDC) studies. In addition, the reported findings have important conservation implications for crayfish, and perhaps for other groups of freshwater organisms. The authors also included fragmented stream habitats in their study and integrated this habitat feature in their experimental design: this represents an additional value, since this condition is pretty common in riverine habitats and could potentially represent a “confounding factor” in SGDC. If the authors will improve scientific language following suggestions, provide convincing answers to the raised comments and work on some minor issues (see Additional comments), this work would deserve publication.

2. The underlying data and additional information have been provided, they are robust and results are statistically sounds.

3. Results are discussed in a neutral way, without any kind of overemphasis; conclusions are linked with the results and are in general well stated (but see Additional comments for some minor points).

Additional comments

# Abstract

Line 29: "each of the three diversity metrics". Not clear to me. Are you referring to diversity "metrics", or levels? Or maybe you are talking about the tested diversity correlations? Please clarify (and better to mention these "three diversity metrics" in brackets).
Moreover, replace "vs" with "and" (between...and)

Line 39-40: replace "reliably" with "indiscriminately" and add something like "without empirically testing this correlation in the focal group" at the end of sentence.

Line 40-42: this last sentence can be improved a little bit: please try to rewrite it in a more scientific way, stressing the importance to assess levels and drivers of community and genetic diversity,
and mentioning the importance of preserving eco-evolutionary processes under a conservation perspective.

# Introduction

Lines 53-55: what you are mentioning here is one of the key concept of the theory of island biogeography, therefore you should add an appropriate citation (e.g., MacArthur & Wilson 1967 book). Moreover, citation is needed for the second part of the sentence where you talk about the consequences of gene flow and drift on genetic diversity. Lastly, I'm aware that parallel effects of eco-evolutionary processes on colonization/gene flow and extinctions/allele loss have been first speculated as primary cause for positive SGDC in seminal papers, but these are not the only forces acting in community ecology, for example interspecific competition/facilitation or environmental filtering and selective pressures can play a major role on a more local scale; see the more recent book by Mark Vellend: The Theory of Ecological Communities (2016) for a comprehensive overview. You discussed some of these aspects later on, so no need to detail everything here, but I think that this incipit, in its actual form, is a bit misleading. Perhaps, it could be more convenient for you to start describing how the field of SGDC study was born, and in this perspective the seminal paper by Vellend & Geber is an appropriate citation.

Lines 60-61: replace "is plausible" with "can occur".

Lines 61-64: the whole sentence is a bit confused. Add “at the community level” before “species richness” at line 61. Replace “ enhanced” with “high”. Replace "feedback between" with "effects of". Add “stochastic” before “ecological”: you are forgetting to mention stochastic processes, which can be important in small populations. Moreover, some citations are needed at the end of the sentence.

Lines 66: replace “possess” with “be characterized by”.

Lines 71 replace "by data" with "by a meta-analysis"; if possible also add a more recent citation.

Line 73: replace “that drive” with “preventing the”; add comma before “preventing”.

Line 74: add “natural” to “selection”; replace “plays” with “may play”.

Line 76: add “in different group of organisms” after “AGDCs”.

Line 81: replace “compilations” with “reviews”.

Line 83: perhaps you could also have a look at this more recent meta-analysis: Xie et al., 2021. A meta-analysis indicates positive correlation between genetic diversity and species diversity. Biology, 10(11), 1089.

Line 90: and what about the potential negative effects of interspecific competition on the genetic diversity of some species (you mentioned also this possibility in the Discussion).

Line 91: replace “may operate independently” with “may not operate in parallel ways”; add “and groups of organisms” at the end of sentence.

Line 96: perhaps replace “physiological tolerances” with a more general “ecological optima”; you might also mention the effects of phenotypic plasticity (which can be quite different in different species).

Line 99: “on some diversity and abundance factors”: not clear to me (particularly the use of the term “factor”).

Line 99: add “in” before “species”.

Line 100: add a comma before “while”; add “in most communities”.

Line 101: add “are” before “often”.

Line 102: add “field and” before “labor”.

Line 103: replace “to surveying and summarizing diversity” with something like “aimed at investigating different levels of biodiversity, as well as their evolutionary and ecological drivers”.

Line 104: add “important” or “crucial” to put some emphasis.

Line 107: replace “population genetics” with “genetic diversity”.

Line 108: replace "make reasonable predictions about the two remaining diversity measures" with "make reasonable predictions about the other two".

Line 110: replace “would” with “information could be used to”

Line 111: add “in biodiversity monitoring” (or “assessment”) after “prioritized”.

Line 113: add “in such situations” before “there is potential”.

Line 115: replace “protection efforts” with “conservation strategies”. Here, it could also be worth to explain the risk of wrong assumptions: when there is no correlation between species and genetic diversity, taking one as a proxy for the other will lead to sub-optimal or detrimental conservation strategies.

Line 119: add “suitable and somehow” before “tractable”; replace “assessing evidence for” with “empirically testing”.

Line 126: “block sections”: I understand what you mean, but not sure if the verb “block” is appropriate here. In an ecological and biomonitoring context, shouldn’t “isolate sections” be better?

Line 135: add “and connectivity” after “habitat”

Line 139: “a decoupling of these diversity measures” sounds bad. Maybe better something like: “a decoupling of the effects on these two diversity levels” (or simply “contrasting effects on..”).

Line 141: add “of populations” after “genetic diversity”.

Line 144: I would prefer “group of organisms” here, instead of the term“taxa”, which have a more taxonomic meaning

Line 151: “ecosystem engineers”: could you briefly explain why?

Line 156: “such those proposed by Taylor et al.”: if you put stress on some specific conservation strategies, you should specify in what they consist (which are their peculiarities?). Otherwise, simply add “effective” to “conservation strategies” and cite the paper as an example.

Line 167: “mitochondrial DNA sequences, as well as complementary nuclear genetic maker data”: please specify mtCOI and ISSR.

# Materials & Methods

Line 199: replace “pattern of sampling site placement” with “sampling design”.

Line 202: replace “Ultimately, there where” with “Overall, our study included”.

Line 206: for the title, I would prefer something like “Crayfish monitoring and sampling” (you did both).

Line 209: I don’t understand why you are citing Barnett et al. 2022 here. Did you rely on this previous study for the collection of samples? Otherwise, what does the citation refer to? Please clarify, and if your study relies on data from previous work, this must be more clearly stated.

Line 223: “number of each species”: did you counted the number of species or the “number of individuals for each species”? Please correct.

Line 228: “the two focal species…”: I think you should spend a couple of words for explaining why you selected these two species (here or in the Introduction). Are they both common and easy to sample? do they differ for some ecological requirements or other biological features?

Line 331: please provide the total number of individuals collected per species.

Line 239: close bracket after “Barnett et al. (2020”; open bracket before “Supplemental Data”.

Line 244: replace “informative” with “used”.

Line 246: you outlined the difference between the two markers, please specify better the implication of this difference: could data from mtDNA and nuc ISSRs provide information on evolutionary processes acting at different temporal scale? (i.e., past vs current processes; keep in mind also the likely difference in mutational rate). The combined use of nuc and mtDNA marker is an interesting aspect of your study, but you didn’t develop it.

Line 271: maybe “operators” could be better than “people”.

Line 274: perhaps simply “Habitat characterization” could be more concise and elegant as title.

Line282-284: not clear to me.

Line 289: replace “captured” with “detected”.

Line 292: "electrofishing and kick seining data were characterized as the number of crayfish individuals collected/100 m 2 and abundances were summed for each reach": please re-write in a cleared way. E.g., “we counted the no. of crayfish collected after electrofishing…”.

Line 308: replace “uniting” with “constructed based on the sequences of”; replace “present within a location” with “found at a certain location”. Please also note that “taxa” and “haplotypes” are not the same things: the term taxa has a strict taxonomic meaning.

Line 311: add “the” before “use”.

Line 315: replace “containing” with “including”.

Line 331: “population genetic diversity”: here and throughout the manuscript, use “within-population genetic diversity”, or simply avoid to specify “population” (implicit).

Line 332: better something like: “we investigated SGDC assessing the relationships between species richness and each of the four…”

Line 336: citation needed (R Core Team 2022).

Line 337: replace “local sites from which F. validus and F. erichsonianus had been sample” with simply “sampling sites for F. validus and F. erichsonianus”.

Line 343: maybe better “relationships” than “association”.

Line 345: perhaps, “habitat characteristics” better than “environmental characteristics” (cause for example, you didn’t measure any climatic variables).

Line 349: replace “If relationships between stream channel characteristics and diversity metrics differ” with something like “If stream stream channel characteristics affect the two levels of diversity in a different way…”.

Line 352: “no correlation”: always no correlation? Are you sure? Couldn’t the mentioned example also lead to negative correlations in some cases?

Lines 374-375: the way you reported the concept is not clear.

Line 376: replace “maintain overall p-value at p = 0.05” with “correct for the effect of multiple comparison on the family-wise error rate”.

Results

Please add a reference to Supplemental Data S2 for COI raw data; where are ISSR raw data?

Lines 385-387: where do the numbers 0.282, 0.723 and 0.016 came from? I don’t see these values in Table 2).

Line 388: please start with “For F. validus, we successfully sequenced 143 individuals, obtaining a 618-bp…”; same for the other species in the next sentence.

Line 389: again, the Barnett et al. 2022 citation: if your study relies on sampling and data from previous work, this must be more clearly stated (see my previous comment about sampling).

Line 395: perhaps simply “Habitat characterization” could be more concise and elegant as title.

Lines 396-398: the reported numbers for LWD and % vegetation do not correspond to the values reported in Table 3. Maybe in Table 3 you only reported the average values? In any case, you cannot mention table 3 after these values, if they are not reported in the table.

Line 400: the title “Diversity metrics” is inappropriate. You are reporting estimates. Something like “Species and genetic diversity estimates”could be better.

Line 401: replace “At the community-level” with “Regarding community-level diversity”, for the sake of clarity. Same at line 402 for population-level diversity (and specify you are talking about genetic diversity).

Line 402: add “mtCOI” before “haplotypic”.

Line 403: correct “was” with “were”.

Line 404: “PD”: better to recall the meaning of acronym the first time you mention it in the results; same for “PPL” at line 406. General comment: you reported only mean values. Please describe variation among sampling sites and also report min/max values (or SD).

Line 411: you should mention the statistic test when reporting the result; e.g., “Pearson correlation test showed no significant correlation between species richness and…”.

Line 414, 418, 422, 433 and 437: same as above.

Line 448: correct “population” in “within-population”, or simply delete it (implicit).

# Discussion

Line 452: I think you should start the Discussion recalling that you found no significant SGDCs in your study system. From a conservation planning perspective (i.e., the opportunity to use one level of diversity as a proxy for the other), this is perhaps the most general and salient finding of your study.

Line 453: at the end of the sentence you must specify something like “although only the first was found to be statistically significant”.

Line 466: add “in AGDC” after “negative trends”.

Line 473-478: ok, that’s interesting and nice discussion. However, why no AGDC was found for nuclear markers? Could you please try to discuss some potential explanation for that? (or at least draw some hypotheses). Keep in mind the different nature of mtDNA vs nuc markers. E.g., could dispersal be sex-biased in crayfish, or could you rule out this hypothesis as potential explanation? Considering that mtDNA marker most likely has the capability to also capture more ancient processes, could your sampling sites in downstream sections capture long-distance dispersal patterns, potentially from tributaries with different evolutionary history? Did the haplotypes detected at downstream sites belong to different haplogroups, or not?

Line 491-492: competition...maintaining genetic diversity. What do you mean? Interspecific or intraspecific competition? Please not that inter-specific competition can reduce genetic diversity in some species by limiting pop size (demographic effects). From a theoretical point of view, inter-specific competition could also promote genetic diversity, by favoring functionally diverse genotypes that allow the species to explore more niche space (to avoid competition). But how about your genetic diversity estimates? Do they come from neutral or adaptive markers? Please clarify all these aspects (or delete this sentence).

Line 511: add “and the environmental context” after the dot.

Line 512: “differ” from the rest of the community in what? And how this could affect SGDC? Please clarify.

Line 513: please re-write this sentence, for example “In this study, we selected as focal species for genetic diversity assessment…”.

Line 524-525: ok, this is a reasonable explanation. But what about potential differences in the evolutionary history of the two species, potentially affecting their genetic diversity pattern? (remember COI is a mitochondrial marker).

Line 528: if you need a more recent citation, have a look at: Quadroni et al., 2016. Effects of sediment flushing from a small Alpine reservoir on downstream aquatic fauna. Ecohydrology.

Line 540: before “the need remains for simple…”, perhaps you should specify something like “in conservation practice” (or planning). Indeed, for other more specific purposes, e.g., the study of adaptive genetic variation, genomic approaches are preferable.

Line 542: replace “genomic variation can be highly repeatable” with “genetic variation can lead to highly reproducible results”.

Line 546: “exiting mtCOI datasets”: I’m sure mtCOI datasets are exiting for you, but perhaps you wanted to write “existing”? : )

Lines 547-548: this sentence is too general. Again, you should spend a paragraph discussing the nature of your markers (mtCOI and nuc ISSR), highlighting their differences and how this could potentially affect SGDC results.

Line 549: delete “conservation” before “implications for the conservation” (to avoid repetition).

Line 550: specify “detected in one species” after “AGDC”.

Line 551: replace “a strategy” with “separate strategies” and delete “separately” at the end of line.

Line 559: replace “metrics” with “diversity”.

Line 560: replace “warrant efforts” with “deserve conservation priority”.

Line 561-562. In your study, you put much efforts not only in assessing patterns, but also in investigating the underlying drivers (i.e., processes). I really think this is a valuable point of your work: perhaps you should highlight it (here or in the Conclusions). Preserving evolutionary and ecological processes (not only patterns) is crucial for the long-term conservation of species, particularly in the face of habitat alteration/fragmentation and environmental change.

# Conclusions

Line 565: for the sake of clarity, maybe you should make explicit the meaning of the acronyms, the first time you mention them in the Conclusions.

Line 567: add “(AGDC)” after “genetic diversity”.

Line 572: change “altered” into “affected”.

Line 576-577: You discuss this at line 582 and following. Avoid redundancy. The more concise = the more effective, particularly in the Conclusions.

Line 577-579: again, this is in part redundant with lines 572-575. Try to merge the paragraphs.

Line 583: I would write something like “eco-evolutionary processes” instead of “environmental factors”, cause this include everything.

Line 586: replace “decrease” with “lead to a loss of” and add something like “with detrimental consequences for the species in the long term”, at the end of the sentence.

Line 587: replace “environmental conditions” with “habitat diversity and integrity”.

Line 592: perhaps add also “and in different riverine ecosystems” at the end of sentence; replace “diversity metrics” with “biodiversity levels”.

Line 593: in the Introduction, you wrote that crayfish are often considered ecosystem engineers: perhaps you could briefly recall the concept here, to further stress the importance of preserving crayfish communities.

Reviewer 2 ·

Basic reporting

no comment

Experimental design

My main concern is about the use of ISSR genetic markers. Please, see in additinola section for details.

Validity of the findings

no comment

Additional comments

The paper entitled "Relationships between crayfish population genetic diversity, species richness, and abundance within impounded and unimpounded streams" is an interesting paper which proposes to relate different parameters of diversity, species abundance, genetics, and environmental characteristics. The paper is well constructed, well documented, and well explained. Apart from various comments reported afterwards, my main concern is the use of ISSRs. I consider that the number of markers is too low and individuals per site is also low, which translates into a very low number of observed bands that will be used to define the quality and robustness of the results. This translates into the observation of no convincing results with these markers. This final result could come from the markers, but increasing the number of markers could change the results. ISSRs are very good markers for assessing genetic diversity, however, some precautions are necessary, such as ensuring a certain number of individuals and markers. My proposal would therefore be either to increase the number of markers or to eliminate this part, which is technically weak anyway, with no results. I think that presenting the paper with just the COI would be just as good.

Specific comments

Abstract section

Line 25-26. “using nuclear (nDNA) and mitochondrial DNA (mtDNA) markers”.
It's not really informative to put nDNA for nuclear and mtDNA for mitochondrial, which is pretty obvious. It's preferable to give information directly on the markers used, i.e. ISSR for the former and COI for the latter.


Introduction section

Introduction is very well structured and complete.


Method section

• Line 230: “On average, sample sizes were six individuals per species per site (SD = 1.12) (Table 1).”
At first glance, it's a bit confusing because the values are presented in the table by stream and not by site. It might be a good to mention the rank of the values for the number of individuals collected by stream and between brackets mention the average, i.e. by site.

Fig. 1. It is not clear which black mark on the principal map represents which drainage. It would help to add some indication (text, arrow, or other).

Table S2. It would help to include a column with the haplotype code because you have to go to genbank to see the haplotype number. A codification between the species name and the haplotype number could help a lot (type: FV_H01, FV_H02 etc) for discussion.
In this same table, it might be good to add a line above indicating the drainage.

ISSR: The number of ISSR markers per species for the study is very low (only 2), and the total number of bands/loci is low (35 and 24). This low number could affect the reliability of the results. It would be good if the authors could justify this low number. For your information, I'm providing 2 publications that may help.
• Mariette, S.; Le Corre, V.; Austerlitz, F.; Kremer, A. Sampling within the genome for measuring within-population diversity: Trade-offs between markers. Mol. Ecol. 2002, 11, 1145–1156.
• Nelson, M.F.; Anderson, N.O. How many marker loci are necessary? Analysis of dominant marker data sets using two popular population genetic algorithms. Ecol. Evol. 2013, 3, 3455–3470.

Electrophorese: the authors mention 16h40 for electrophoresis at 100 Volts. Are you sure about this? It seems really long. We get good separation results for ISSR on a 2% gel at 80 volts for about 4h-4h30 at room temperature.

Line 332: For the SGDC evaluation, it is not clear at what scale this is done. Is it by site, by stream, by drainage? Please, specify.

Line 372: At this level of reading, it's not very clear how RIV is determined. Perhaps later on it will become clearer, but it would be good to give a little more explanation at this stage of the manuscript.


Results section

Table 2: total values for the Cahaba River are in bold, while those for the Bear Creek are not. Please, homogenize.

Line 391. “We obtained ISSR data from 109 F. validus and 95 F. erichsonianus individuals”.
It would be good if the authors could explain why all the individuals tested for COI were not tested with ISSRs.

Line 396-398: In the table of these results, the authors refer to a D50 value, but this is not how the methods are presented. Adjust in the methods. What's more, in the manuscript, the authors refer to the "median", whereas in the table, for the same parameter, they refer to the "mean". Please, clarify. The authors refer to LWD from 2 to 25 pieces, but this is not clear in table 3. Clarify. The same goes for % of vegetation. It might be good to put the minimum and maximum values instead of 1 standard deviation, which would be more consistent with the description in the manuscript; or put the 2 types of values.

Line 401. I can't find the table showing the Chao-1 species richness parameter.

Line 412-415. This can be summarized in a single statement that no significant differences are observed between species and genetic markers.

Line 419. “As such, the MIH was not supported”. Can't the same thing be said for the SGDC? in the previous paragraph?

Table 5. In the legend of table 5, the authors refer to the level of signification of the tests by stars (* or ** or ***), but in the table we don't see this reference (no star).

Line 433: “Species richness was correlated with stream width”. A level of correlation will always be observed even weak, but it's not clear whether the authors are saying that this is significant? This is not clear from table 5. Furthermore, figure 2a shows that the relationship is inverse between impounded and unimpounded stream, but this does not appear in table 5. Table 5 therefore considers the 2 data types together. This part is unclear.


Results from: Association between environmental characteristics, community- level diversity, and population-level Diversity.
It would be good to insist on significant relationships, which in reality are those that can be considered important. the others can be presented and discussed with caution due to the absence of significant power. I insist on the fact that it is not clear how the RVI is determined and therefore how to interpret it.

Figure 2. I'm confused by the scale presented for median substrate size compared to the results in table 3.


Discussion section

General comment: Could there be an impact on the results and therefore on the discussion due to the fact that during the 2016/2017 sampling only the Cahaba river drainage site was sampled and only individuals of the F. erichsonianus species were collected. Doesn't this introduce a geographical and temporal bias?

line 541-542: “Sinn et al. (2022) demonstrated that using ISSRs to assess genomic variation can be highly repeatable.”
While it's true that Sinn et al mention this, they are considering a technique derived from ISSRs that consists of sequencing a group of ISSRs, which does not correspond to the method defined in the manuscript. I think there are many papers that highlight the repeatability of ISSRs as genetic markers and their effectiveness in determining genetic diversity. I invite the authors to better target the reference in this case.


Line 549: “Our findings have several important conservation implications for the conservation of crayfish diversity.”
Eliminate the first word “conservation”. Then sentence is: Our findings have several important implications for the conservation of crayfish diversity.”

Style comment section

General: Multiple references to the same item should be separated with a semicolon (;) and ordered chronologically. Please adjust with semicolon (as mentioned in Reference format of the manuscript guidelines).

Line 64: (e.g., Allee effects, and inbreeding). No coma

Line 483. “In our study we found that stream”. Coma after study.

Line 552: “Here we showed that fragmentation changed the relationship”. Coma after Here.

·

Basic reporting

The manuscript addresses a relevant issue using an original methodology. The article uses clear English, and the text is technically correct. However, minor corrections are necessary before it can be published in PeerJ.

Experimental design

No comment

Validity of the findings

No comment

---

## Round 0.2 · accepted · Accept

Thank you for the revision of the manuscript. I hereby certify that you have adequately taken into account the reviewer's comments and improved the manuscript accordingly. Based on my assessment as an Academic Editor, your manuscript is now ready for publication. The editorial comments of reviewer 2 can easily be considered in the correction phase of the proofs.

·

Basic reporting

The authors have carefully taken into account all of my comments, implementing the suggested changes to the manuscript or providing detailed and convincing replies. Inaccurate and unclear points have now been resolved and the manuscript has been significantly improved. Therefore, I now recommend the manuscript for publication. A few minor comments, mainly related to grammar and language style, are listed in: “Additional comments”.
1. The manuscript is overall well-written concerning language and style.
2. Introduction and background are appropriate, as well as literature citations.
3. The article has a rigorous structure and follows the PeerJ format.
4. Figures and tables are relevant and well described.
5. All the raw data have been provided and reproducibility is granted

Experimental design

1. Original research is within the scope of the journal.
2. Research questions are well defined and addressed in detail; the tested hypothesis framework is complex and carefully designed, with careful considerations of all the potential underlying factors.
3. Field and lab works are considerable; all the analyses were performed rigorously.
4. Methods are well described and reproducibility of analyses is granted.

Validity of the findings

1. This study represents a valuable contribution in the context of species-genetic diversity correlation (SGDC) studies. In addition, the reported findings have important conservation implications for crayfish, and perhaps for other groups of freshwater organisms. The authors also included fragmented stream habitats in their study and integrated this habitat feature in their experimental design: this represents an additional value, since this condition is pretty common in riverine habitats and could potentially represent a “confounding factor” in SGDC.
2. The underlying data and additional information have been provided, they are robust and results are statistically sounds.
3. Results are discussed in the proper ways; conclusions are linked with the results and are well stated.

Additional comments

Note: I will refer here to line numbers in the manuscript without track changes.
Line 94: replace "and" with "and/or": not always both the opposite influence of environmental drivers and competition act together; perhaps also add "interspecific" before "competition", if you are referring to that kind of competition.
Line 409: maybe "If there WERE" instead of "If there was"?
Line 501: "seem", not "seems"
Line 508: here and in the following lines you frequently make use of the word "systems",
but the results of your studies specifically refer to riverine systems (and do not necessarily extend to other ecosystems). At least once, at the beginning of this paragraph, it's better to specify that, simply adding "riverine" before "systems".
Line 510: "deserve conservation priority to conserve genetic diversity". To avoid repetition "conservation...to conserve", you could write "prevent the loss of" (or "counteract the loss of") instead of "conserve".

Reviewer 2 ·

Basic reporting

without comment

Experimental design

without comment

Validity of the findings

without comment

Additional comments

Line 396-398. Authors did not adjust table considering comment. They say 2-25 pieces but only put the median. I ask for min and max but they still with SD. Maybe they can refer to Table S5 for detail.

Line 477: with stream width. Repeat twice, I suppose that is an error.